# The G3BP1-UPF1-Associated Long Non-Coding RNA CALA Regulates RNA Turnover in the Cytoplasm

**DOI:** 10.3390/ncrna8040049

**Published:** 2022-06-30

**Authors:** Luisa Kirchhof, Youssef Fouani, Andrea Knau, Galip S. Aslan, Andreas W. Heumüller, Ilka Wittig, Michaela Müller-McNicoll, Stefanie Dimmeler, Nicolas Jaé

**Affiliations:** 1Institute of Cardiovascular Regeneration, University Hospital, Goethe University Frankfurt, 60590 Frankfurt, Germany; kirchhof@med.uni-frankfurt.de (L.K.); youssef.fouani@gmail.com (Y.F.); knau@med.uni-frankfurt.de (A.K.); aslan@med.uni-frankfurt.de (G.S.A.); andreas.heuhk@gmail.com (A.W.H.); dimmeler@em.uni-frankfurt.de (S.D.); 2Faculty of Biological Sciences, Goethe University Frankfurt, 60438 Frankfurt, Germany; 3German Center of Cardiovascular Research (DZHK), 60590 Frankfurt, Germany; wittig@med.uni-frankfurt.de; 4Functional Proteomics, Institute for Cardiovascular Physiology, University Hospital, Goethe University Frankfurt, 60590 Frankfurt, Germany; 5Institute of Molecular Biosciences, Goethe University Frankfurt, 60438 Frankfurt, Germany; mueller-mcnicoll@bio.uni-frankfurt.de

**Keywords:** long non-coding RNA, RNA turnover, nonsense-mediated mRNA decay, gene expression, G3BP1

## Abstract

Besides transcription, RNA decay accounts for a large proportion of regulated gene expression and is paramount for cellular functions. Classical RNA surveillance pathways, like nonsense-mediated decay (NMD), are also implicated in the turnover of non-mutant transcripts. Whereas numerous protein factors have been assigned to distinct RNA decay pathways, the contribution of long non-coding RNAs (lncRNAs) to RNA turnover remains unknown. Here we identify the lncRNA *CALA* as a potent regulator of RNA turnover in endothelial cells. We demonstrate that *CALA* forms cytoplasmic ribonucleoprotein complexes with G3BP1 and regulates endothelial cell functions. A detailed characterization of these G3BP1-positive complexes by mass spectrometry identifies UPF1 and numerous other NMD factors having cytoplasmic G3BP1-association that is *CALA*-dependent. Importantly, *CALA* silencing impairs degradation of NMD target transcripts, establishing *CALA* as a non-coding regulator of RNA steady-state levels in the endothelium.

## 1. Introduction

Maintaining equilibrium between RNA synthesis and decay is crucial for cellular function and homeostasis. Following transcription, the fate of a given RNA is directly linked to the protein factors it associates with in a structure and sequence-dependent manner [1]. Functionally, RNA decay serves a dual purpose. While it systemically removes aberrant and potentially toxic transcripts, RNA decay also counteracts transcription by balancing the levels of unmutated mRNA transcripts through degradation. In this way RNA decay enables fast changes in the RNA repertoire in response to external stimuli [2]. Concerning RNA surveillance and quality control, the degradation of defective transcripts is initiated by their identification through specific protein factors (e.g., the exon junction complex) followed by feeding them to the RNA decay machinery [2]. In this context, the nonsense-mediated decay (NMD) pathway, which eliminates mutant transcripts harboring a premature translation termination codon is a major route for RNA degradation [3]. Mechanistically, NMD is directly linked to translation termination [4] and essential to prevent accumulation of C-terminally truncated polypeptides. Key to this process is the ATP-dependent RNA helicase UPF1 (Regulator of nonsense transcripts 1) which catalyzes the remodeling of mutant RNA–protein complexes and is tightly controlled by various means [5,6,7]. Recent studies demonstrated that UPF1 is ~10-fold more abundant than the two other core NMD factors, UPF2 and UPF3X, suggesting that UPF1 might function beyond NMD [8,9]. In agreement with this, UPF1 was demonstrated to be involved in a variety of other RNA decay pathways, including Staufen-mediated, replication-dependent, or glucocorticoid receptor-mediated mRNA decay (extensively reviewed in [8]). Interestingly, other studies emphasized a gene-regulatory role for UPF1 and the NMD pathway. Particularly, the importance of mRNA decay of non-mutant transcripts was recently highlighted [10] and this process was demonstrated to be dynamically regulated by developmental and environmental cues [11]. Besides distinct molecular pathways, subcellular membrane-less ribonucleoprotein assemblies, like cytoplasmic stress granules (SGs) or processing bodies were shown to play a central role in RNA metabolism and turnover [12]. For the formation of SGs, which are RNA-protein aggregates formed upon exposure to cellular stress, the protein factors Ras GTPase-activating protein-binding protein 1 (G3BP1) and 2 (G3BP2) were demonstrated to be indispensable [13]. Given the high expression levels of these proteins in cellular homeostasis, surprisingly little is known about their roles and functions outside the stress response. While first results point towards a general contribution to RNA metabolism [14], knowledge about their contribution to basal RNA turnover is just emerging.

Here we demonstrate that the long non-coding RNA (lncRNA) *LINC00674* (hereafter named *CALA*: cytoplasmic G3BP1-associated lncRNA) participates in RNA turnover control in the endothelium via interaction with G3BP1-positive ribonucleoprotein (RNP) complexes. In contrast to other lncRNAs, which impact RNA decay in the nucleus [15] or engage in cytoplasmic RNA turnover via direct lncRNA-mRNA interactions [16,17], *CALA* regulates mRNA turnover in the cytoplasm by enhancing the local concentrations of NMD factors required for RNA decay. Specifically, combining antisense affinity selection, mass spectrometry, and RT-qPCR, we revealed that *CALA* is part of cytoplasmic RNPs containing G3BP1 in addition to well-known G3BP1 interaction partners, such as G3BP2, CAPRIN1, PABPC1, and UPF1. In line with this, we showed that *CALA* silencing under homeostatic conditions impairs the formation of these cytoplasmic RNPs and leads to significantly increased expression levels of known non-mutant NMD targets. In summary, this study established the lncRNA *CALA* as a novel non-coding regulator of RNA turnover in the endothelium.

## 2. Materials and Methods

### 2.1. Lead Contact

Further information and requests for resources and reagents should be directed to, and will be fulfilled by, the lead contact, Dr Nicolas Jaé (jae@med.uni-frankfurt.de).

### 2.2. Materials Availability

This study did not generate new unique reagents.

### 2.3. Cell Culture and Gene Silencing

Pooled human umbilical vein endothelial cells (HUVECs; Lonza, Basel, Switzerland; Promocell, Heidelberg, Germany) were cultured in endothelial basal medium (EBM), supplemented with EGM SingleQuots (Lonza, Basel, Switzerland) and 10% FCS (Invitrogen, Waltham, MA, USA) at 37 °C and 5% CO_2_. For stimulation, HUVECs were either exposed to ER stress (1 mM DTT, 24 h), hypoxia (0.2% O_2_, 5% CO_2_, 24 h) or nutrient depletion (reduced-serum media (Opti-MEM with GlutaMax Supplement, Thermo Fisher, Waltham, MA, USA), 24 h). HeLa cells were cultured in DMEM (Thermo Fisher, Waltham, MA, USA) supplemented with 10% FCS (Invitrogen, Waltham, MA, USA) and 1% Penicillin/Streptomycin at 37 °C and 5% CO_2_. All cells were regularly tested negative for mycoplasma.

For silencing of gene expression, cells were transfected with LNAs (50 nM) using Lipofectamine RNAiMax (Life Technologies, Waltham, MA, USA) according to the manufacturer’s instructions.

### 2.4. RNA Isolation, RT-qPCR, and RT-dPCR

Total RNA from HUVEC and HeLa cell culture was isolated and DNase digested using RNeasy Mini Kits (Qiagen, Venlo, The Netherlands), according to the manufacturer’s instructions, and quantified with a NanoDrop 2000 spectrophotometer (Thermo Fisher, Waltham, MA, USA). cDNA synthesis was done from 500 ng RNA, using random hexamers and M-MLV reverse transcriptase (Thermo Fisher, Waltham, MA, USA). RPLP0-normalized (2^−ΔCt^) quantitative (q) PCR reactions were performed on StepOnePlus real-time PCR cyclers (Thermo Fisher, Waltham, MA, USA) and digital (d) PCR reactions were run on a QIAcuity One system (Qiagen, Venlo, The Netherlands).

### 2.5. Oligonucleotides

Primers were purchased from Sigma-Aldrich (St. Louis, MO, USA), LNAs from Qiagen (Venlo, Netherlands), and desthiobiotinylated 2′O-Me-RNA probes from Integrated DNA Technologies (IDT, Coralville, IA, USA). All sequences are listed in Appendix A.

### 2.6. In Vitro Sprouting Assays

HUVEC spheroid sprouting assays were conducted as described elsewhere [18]. Briefly, cells were detached after 24 h of transfection and seeded in an EBM-methocel mixture (80:20) to 96-well U-bottom plates to allow spheroid formation by incubation for 24 h at 37 °C. Successfully formed spheroids were resuspended in a methocel-FCS mixture (80:20) before the same amount of rat-tail collagen type I (Corning Inc., Corning, NY, USA) was added. The spheroids were embedded by plating to 24-well plates and incubated for 24 h at 37 °C under basal conditions or presence of VEGFA (50 ng/mL). The next day, pictures of 10 spheroids per condition were taken using an Axio Observer Z1.0 microscope (Zeiss, Oberkochen, Germany) at 10× magnification. The cumulative sprouting length was determined using the Zeiss AxioVision digital imaging software (version 4.6).

### 2.7. Migration Assays

To determine the migratory capacity of HUVECs, transfected cells were cultured in fibronectin-coated 2-well cell culture inserts (Ibidi, Gräfelfing, Germany) for 24 h. By removing the cell culture inserts a cell-free gap between two confluent cell layers was created and gap closure was subsequently recorded by taking pictures at the indicated time points using an Axio Observer Z1.0 microscope (Zeiss, Oberkochen, Germany). Gap closure was analyzed with respect to the starting points.

### 2.8. Cell Proliferation

Cell proliferation was assessed by incorporation of BrdU using the BrdU Flow Kit (BD Biosciences, Franklin Lakes, USA), according to manufacturer’s protocol. Briefly, transfected HUVECs were incubated with BrdU (3.1 µg/mL, 45 min, 37 °C) and successively washed with PBS, Cytofix/Cytoperm buffer, Perm/Wash buffer, and Cytoperm/Permeabilization buffer plus. Thereafter, cells were incubated in DNase I (1 h, 37 °C), washed again with Perm/Wash buffer, and incubated with V450 mouse anti-BrdU antibodies (clone 3D4, 20 min, RT). Finally, 7-AAD was added (10 min, RT) and cells were analyzed, using a FACS Canto II device and the FACSDiva software (BD Biosciences, Franklin Lakes, NJ, USA).

### 2.9. Apoptosis

Apoptosis rate was determined as caspase-3/-7 activity using Caspase-Glo^®^ 3/7 assay (Promega, Fitchburg, MA, USA), according to manufacturer’s protocol. To this end, transfected HUVECs were incubated for 1 h at 37 °C with diluted Caspase-3/-7 substrate in Caspase Glo buffer and caspase-3/-7 activity was determined via luminescence measurement using a GloMax-Multi+ Detection System (Promega, Fitchburg, MA, USA).

### 2.10. In Vitro Permeability Assays

To analyze the permeability of endothelial cells, HUVECs were cultured on fibronectin-coated cell culture inserts (ThinCert, 1 µm pore diameter, 24-well, Greiner Bio-One, Kremsmünster, Austria) to create a confluent cell layer and incubated for 1 h at 37 °C with FITC-dextran containing media (70 kDa, 1 mg/mL, Sigma-Aldrich, St. Louis, MO, USA) and the extravasation of FITC-dextran through the endothelial monolayer was analyzed by measuring the fluorescence (λex = 493 nm, λem = 518 nm) in the lower chamber using a GloMax-Multi+ Detection System (Promega, Fitchburg, MA, USA).

### 2.11. Sucrose Density Gradient Ultracentrifugation

HUVECs were lysed in 50 mM Tris-HCl (pH 8.0), 50 mM NaCl, 0.5% NP-40, incubated for 30 min on ice. Cell lysates were cleared by centrifugation and further treated with proteinase K (0.005 mg/µL, VWR, Radnor, PA, USA) and RNase inhibitor (4 U/µL, Thermo Fisher, Waltham, MA, USA) for 30 min at 37 °C. For mock treatment, proteinase K was substituted with nuclease-free H_2_O. After dilution with 5% sucrose solution (5% (*w*/*v*) sucrose, 20 mM Tris-HCl pH 8.0, 5 mM MgCl_2_, 100 mM KCl, protease inhibitor cocktail (Roche, Basel, Switzerland)), lysates were loaded on top of a 4 mL 15–55% sucrose density gradient and centrifuged for 2 h 30 min at 200,620× *g* at 4 °C using an MLS-50 rotor (Optima MAX-XP ultracentrifuge, Beckman Coulter, Brea, CA, USA). Per gradient, 14 fractions were taken, total RNA was isolated and cDNA synthesis was performed as described above.

### 2.12. RNA Antisense Affinity Selection

RNA antisense affinity selection was performed as mentioned [18,19]. For the antisense affinity selection of endogenous *CALA*-protein complexes, HeLa cells were lysed in 50 mM Tris-HCl (pH 8.0), 50 mM NaCl, 0.05% NP-40 (*v*/*v*), supplemented with protease inhibitor cocktail (Roche, Basel, Switzerland) for 30 min on ice. After centrifugation for 3 min at 16,000× *g* at 4 °C, cleared lysates were adjusted to 1 mL with 10× RNase H buffer (500 mM Tris-HCl (pH 8.0), 750 mM KCl, 30 mM MgCl_2_, 100 mM DTT, RiboLock (Thermo Fisher, Waltham, MA, USA)) and incubated with 100 pmol non-targeting control or *CALA*-targeting probes carrying 3′ desthiobiotin-TEG for 2 h at 4 °C. Subsequently, *CALA*-protein complexes were captured by ON incubation with DynaBeads MyOne Streptavidin C1 beads (Thermo Fisher, Waltham, MA, USA) at 4 °C. Beads were washed twice with washing buffer (50 mM Tris-HCl (pH 8.0), 150 mM NaCl, 0.05% NP-40 (*v*/*v*), 1 mM EDTA) and once with the same buffer lacking NP-40. Finally, beads were biotin-eluted at 37 °C for 30 min and eluates were subjected to RT-qPCR analysis and mass spectrometry.

### 2.13. Mass Spectrometry

The mass spectrometry data sets, experimental details, and statistics have been deposited with the ProteomeXchange Consortium via the PRIDE partner repository [20] and are publicly available with the data set identifiers PXD033516 and PXD033517.

### 2.14. RNA Immunoprecipitation

For immunoprecipitation of RNA-protein complexes, 50 µL Dynabeads Protein G (Thermo Fisher, Waltham, MA, USA) were washed with 1× IP buffer (50 mM NaCl, 75 mM KCl, 2.5 mM MgCl_2_, 0.5 mM DTT, 50 mM Tris-HCl (pH 8.0), 0.05% NP-40 (*v*/*v*)), incubated with 6 µg antibody of interest (G3BP1 (Cat# 13057-2-AP; Proteintech, Rosemont, CA, USA), hnRNPH1 (Cat# ab10374; Abcam, Cambridge, UK), IgG (Cat# 12-370, Millipore, Darmstadt, Germany)) ON at 4 °C and thoroughly washed again with the same buffer. Meanwhile, HUVECs were lysed using NE-PER Nuclear and Cytoplasmic Extraction Kit (Thermo Fisher, Waltham, MA, USA), according to manufacturer’s instructions. Depending on the experimental design, resulting cytoplasmic and nuclear extracts were either pooled or kept separately. Regardless, extracts were adjusted to 2 mL using 2× IP buffer (100 mM NaCl, 150 mM KCl, 5 mM MgCl_2_, 1 mM DTT, 50 mM Tris-HCl (pH 8.0), 0.05% NP-40 (*v*/*v*), 2× protease inhibitor (Thermo Fisher, Waltham, MA, USA)) and 80 U RiboLock (Thermo Fisher, Waltham, MA, USA) were added. Next, 50 µL beads were resuspended in 1 mL lysate, incubated ON at 4 °C and thoroughly washed with 1× IP buffer. After washing with 1× PBS, beads were used for analysis by RT-qPCR, western blot and mass spectrometry.

### 2.15. SDS-PAGE and Western Blot

HUVECs were lysed in 1× RIPA buffer (Sigma-Aldrich, St. Louis, MO, USA) supplemented with protease and phosphatase inhibitors (Thermo Fisher, Waltham, MA, USA) for 30 min on ice. Next, protein concentrations of cleared lysates were determined by Bradford assays and equal amounts were separated by SDS-PAGE followed by blotting to 0.45 µm nitrocellulose membranes (Cytiva, Marlborough, MA, USA). For the analysis of RIP samples, equal sample volumes were separated by SDS-PAGE. After blotting, membranes were blocked by incubation with 5% BSA or 5% milk powder in 1× TBS-T for 1 h at RT and incubated with antibodies detecting G3BP1 (13057-2-AP (1:5000 in 5% milk powder); Proteintech, Rosemont, CA, USA), hnRNPH1 (ab10374 (1:5000 in 5% milk powder); Abcam, Cambridge, UK) or GAPDH (14C10, #2118 (1:1000 in 5% BSA); Cell Signaling Technology, Danvers, MA, USA) ON at 4 °C. Membranes were thoroughly washed with 1× TBS-T and Incubated with HRP-conjugated anti-rabbit secondary antibodies (1:5000 in blocking solution, GE-Healthcare, Chicago, IL, USA) for 1 h at RT. Finally, membranes were developed using a chemiluminescent HRP substrate (Immobilon HRP substrate, Merck Milipore, Darmstadt, Germany) and imaged using a ChemiDoc Touch imaging system (Bio-Rad, Hercules, CA, USA).

### 2.16. Cellular Fractionation

Cellular fractionation was done using NE-PER Extraction Kits (Thermo Fisher, Waltham, MA, USA), according to the manufacturer’s instructions. Briefly, HUVECs were washed and lysed in ice-cold CER I buffer for 10 min on ice. Next, CER II buffer was added, and samples were further incubated for additional 1 min on ice. Following centrifugation (16,000× *g*, 5 min, 4 °C), the cytoplasmic supernatant was taken, and nuclei were lysed in NER buffer for 40 min on ice. The nuclear lysate was cleared (16,000× *g*, 10 min, 4 °C) and RNA was isolated from cytoplasmic and nuclear fractions.

### 2.17. Statistics

Data are presented as means ± SEM and n refers to the number of independent biological replicates. Data normality was assessed using the Shapiro-Wilk normality test. Statistical significance was determined by the two-tailed unpaired *t*-test or Mann-Whitney U test. Multiple comparisons were performed using two-way ANOVA with Tukey’s or Sidak’s correction. Probability values of less than 0.05 were considered significant and indicated as follows: * *p* < 0.05, ** *p* < 0.01, *** *p* < 0.001, **** *p* < 0.0001.

## 3. Results

### 3.1. Stimuli-Responsive lncRNA CALA Regulates Endothelial Sprouting and Migration

Located at the interface between the blood and the surrounding tissue, endothelial cells are paramount in sensing, and quickly adapting to environmental changes, thus guaranteeing cellular homeostasis and function [21]. The contribution of lncRNAs to these processes is not well-understood. Searching for stimuli-responsive lncRNAs in published datasets, 7 transcripts caught our interest, which were found to be rapidly activated in response to different external stimuli in numerous cell types [22]. Analyzing the expression levels of these lncRNAs in human umbilical vein endothelial cells (HUVECs), the lncRNA *CALA* was found to be the highest expressed (Figure 1A,B). Absolute quantification of *CALA* levels in HUVECs under basal conditions confirmed a robust expression of ~490 copies/ng total RNA (Figure 1C), identifying *CALA* as a highly expressed, yet uncharacterized, lncRNA. To analyze the role and function of *CALA* in the endothelium, we first assayed expression changes of *CALA* upon different stimuli. Interestingly, ER stress, hypoxia, and starvation stimuli impacting endothelial cell functions [23,24,25] significantly drove *CALA* expression in HUVECs (Figure 1D–F and Appendix A).

Next, we silenced *CALA* expression in HUVECs using locked nucleic acids (LNAs) (Figure 1B) and assessed changes in endothelial cell functions. While transfection of HUVECs with *CALA*-targeting LNAs led to a ~90% decrease in *CALA* levels (Figure 1G), we did not observe effects on cell proliferation, apoptosis, or endothelial permeability (Appendix A). In contrast, *CALA* silencing significantly impaired in vitro endothelial sprouting under basal, as well as under VEGFA-stimulated, conditions (Figure 1H) and, additionally, cell migration was significantly reduced (Figure 1I and Appendix A). Taken together, these results indicated that *CALA* was required for the angiogenic capacity of HUVECs *in vitro*, presumably through contributing to cell migration.

### 3.2. CALA Interacts with Multiple RNPs and Primarily Associates with Cytoplasmic G3BP1

To understand the underlying molecular mechanism of *CALA* function in endothelial cells, we first assessed the coding potential of *CALA* which is located on chromosome 17 and constituted out of 6 exons (Figure 2A). RNA-seq data, ribosome profiling and computed coding probabilities [26] (Figure 2A and Appendix A) show that *CALA* is spliced but does not encode proteins. Using sucrose density gradient ultracentrifugation of mock and proteinase K-treated total cell lysates, we found that *CALA* was strongly complexed with proteins, based on the substantial shift of the *CALA* signal away from the high molecular weight fractions towards the lighter ones (fraction #7 → #2–4) upon proteinase K treatment (Figure 2B). To identify the endogenous protein interactome of *CALA*, we deployed antisense affinity purification of *CALA*-RNPs (Appendix A). To this end, we designed two distinct desthiobiotinylated 2′O-Me-RNA antisense probes (probe #1, probe #2), as described elsewhere [18,19] (Figure 1B), and purified *CALA*-RNPs from whole cell lysates. Purified RNA and protein fractions were subjected to RT-qPCR and mass spectrometry, respectively (Appendix A). We detected a significant enrichment of *CALA* over an unspecific non-target control (NTC) for probe #1 and #2 (Figure 2C). In parallel, mass spectrometry of the co-purified protein fractions enabled the identification of *CALA* binding partners (PXD033516, Figure 2D). In detail, probes #1 and #2 specifically enriched 128 and 85 proteins, respectively, with 71 proteins being significantly enriched by both probes (Figure 2E). In contrast to the *CALA*-specific probes, usage of the NTC control probe did not enrich any proteins (Appendix A) and comparison to our previously published interactome datasets of the lncRNAs *NTRAS* [19] and *GATA6-AS* [18] revealed a *CALA* specific protein interactome (Appendix A).

Using the STRING tool [27] to assess protein function associated networks, we identified the majority of *CALA*-associated proteins as components of “ribonucleoprotein (RNP) complex” (Figure 2F). When further clustering was applied, almost all RNPs fell into two distinct groups: 1. heterogeneous nuclear ribonucleoproteins (hnRNPs), including hnRNP H1, H2, H3, hnRNP F, and the G-rich sequence factor 1 (GRSF1) [28], and 2. “cytoplasmic RNP granule” proteins including G3BP1 and G3BP2 (Figure 2F).

We first validated the interaction between *CALA* and hnRNP H1 as a representative for its family members by RNA immunoprecipitation (RIP) and observed a strong interaction of hnRNP H1 with *CALA* (Appendix A). While hnRNPs are well-known to regulate pre-mRNA splicing [29,30], an initial study we conducted excluded a splicing-regulatory function for *CALA* [19].

For the second identified protein network of cytoplasmic RNP granule proteins, factors involved in mRNA metabolism, were prevalent. Specifically, *CALA* affinity selection co-enriched the multi-functional homologs G3BP1 and G3BP2 (Figure 2D,F). Besides forming homo- and hetero-multimers [31,32], both proteins engage in RNA/DNA binding and numerous protein interactions [33,34]. In line with this, we also enriched numerous G3BP-binding proteins (Figure 2D,F): E.g., CAPRIN1 (Cytoplasmic activation/proliferation-associated protein-1), PABPC1 (Polyadenylate-binding protein 1), NUFIP2 (Nuclear fragile X mental retardation-interacting protein 2), ATXN2L (Ataxin-2-like protein), and UPF1 [34] (Figure 2D,F). Subsequently, we validated the interaction between *CALA* and G3BP1, the key component of *CALA*’s identified cytoplasmic RNP granule protein network, by RIP from total cell lysate which revealed a substantial fraction of *CALA* (~20%) to be bound by G3BP1 (Figure 2G). Since the interactome of *CALA* clearly separates into nuclear and cytoplasmic protein fractions, we next analyzed the subcellular distribution of *CALA* and accordingly found *CALA* present in both compartments, compared to specific marker transcripts (Figure 2H). Subsequently, we specifically addressed the site of interaction between *CALA* and G3BP1 by anti-G3BP1 RIPs in cytoplasmic and nuclear fractions and demonstrated an almost exclusive cytoplasmic interaction between both binding partners (Figure 2I).

Taken together, these results revealed that the cytoplasmic fraction of *CALA* was extensively complexed in multivalent G3BP1-RNPs, the constituents of which are known for their mutual interactions and are functionally related to RNA metabolic processes.

### 3.3. CALA Impacts the Composition and Integrity of Cytoplasmic G3BP1-RNPs Driving mRNA Decay

To analyze the function of *CALA* within the identified G3BP1-positive RNPs, we first performed anti-G3BP1 RIPs in cytoplasmic lysates, followed by mass spectrometry (PXD033517, Figure 3A, left). Strikingly, this analysis revealed an overlap of 68 proteins (40%) that were also recovered by *CALA* antisense affinity selections (Figure 3B, Appendix A), highlighting a shared protein interactome. Among the proteins, we identified the G3BP1 binding partners G3BP2, CAPRIN1, PABPC1, NUFIP2, ATXN2L, and UPF1 [34], strongly supporting the notion that *CALA* is also part of this complex. We next repeated the anti-G3BP1 RIPs in *CALA*-silenced HUVECs and determined changes in the G3BP1 interactome by mass spectrometry (PXD033517, Figure 3A, right). After excluding an effect of *CALA* silencing on G3BP1 expression (Appendix A), we observed 36 of the G3BP1-interacting proteins to be differentially associated with G3BP1 upon *CALA* silencing. While only 3 proteins showed an augmented G3BP1 interaction, the remaining 33 proteins were significantly decreased in their G3BP1 interaction (Figure 3C). Gene ontology analysis assigned the majority of these proteins to NMD [35] (Figure 3C,D and Appendix A). Among the NMD proteins, we found reduced interactions between G3BP1 and UPF1 (Figure 3C,D). UPF1 is a key regulatory factor of NMD and other mRNA decay pathways, thereby broadly affecting mRNA fate (extensively reviewed in [8]). Furthermore, interactions of the RNA helicase MOV10 with G3BP1 were reduced (Figure 3C,D). MOV10 has been reported to be involved in mRNA decay [36], and was shown to be especially important for proper UPF1-mediated mRNA decay, as the translocation of MOV10 on target mRNAs ensures removal of secondary structures and proteins, enabling decay [37]. Additionally, *CALA* silencing disrupted the interaction between G3BP1 and ribosomal proteins like RPS6, RPS8, RPS11, and RPL23A (Figure 3C,D). Interestingly, G3BP1 has been previously shown to interact with 40S subunits [38] and with respect to mRNA decay, UPF1 was reported to interact with 40S subunits upon ribosome stalling and the initiation of NMD [39]. Together, these findings support the notion that *CALA* is involved in NMD.

Given these results and the implication of NMD in RNA decay of non-mutant transcripts, we sought to examine the impact of *CALA* on RNA decay. Therefore, we measured the expression levels of well-known non-mutant NMD targets [40,41] upon *CALA* silencing under homeostatic conditions. Strikingly, *CALA* silencing significantly induced the mature transcript levels of *GADD45A*, *GAS5*, and *RP9P* (Figure 3E–G and Appendix A), while we could not observe changes in their precursor levels (Figure 3E–G and Appendix A). Of note, correctly spliced, non-mutated *GADD45A* and *GAS5* transcripts are generally reported to be key targets of NMD, as their RNA levels must be kept low for preventing apoptosis and growth arrest [42,43]. While we could not detect a profound change in cell cycle progression, the observed impairment of angiogenesis and migration was in agreement with the role of GADD45A [44] and GAS5 [45] in endothelial cells.

In summary, our data indicated that *CALA* is required for the stabilization of cytoplasmic G3BP1-positive RNPs implicated in mRNA decay (Figure 4). We showed that the loss of *CALA* led to the disintegration of G3BP1-RNPs and, ultimately, to the stabilization of target transcripts, as evidenced by an increase in RNA expression levels (Figure 4). Overall, these results describe *CALA* as a regulator of RNA decay in the cytoplasm.

## 4. Discussion

Homeostatic gene expression is tightly controlled and regulated at every step of the RNA life cycle. Besides transcription, processing, and RNA transport mechanisms, cytoplasmic RNA decay constitutes an important mechanism regulating ultimate transcript levels. Several pathways are known to regulate RNA decay rates and among these, NMD plays a pivotal role. Initially discovered as a translation-dependent process aiming to eliminate aberrant mRNAs [46], NMD was recently recognized for its role in non-mutant RNA turnover [47]. Thereby, NMD specifically balances transcriptional rates and enables a quick response to cellular stimuli requiring rapid changes of RNA expression levels [47]. Up to now, NMD has been known to be predominantly regulated via NMD factor availability [48,49], competition with other decay pathways [49,50], and by autoregulatory processes [49,51,52,53].

In our study, we identified the undescribed lncRNA *CALA* to impact NMD efficiency by affecting the stability and composition of cytoplasmic G3BP1-positive RNPs involved in RNA turnover. Deploying antisense affinity selection of endogenous *CALA*-protein complexes from whole cell lysates, we identified *CALA*’s protein interactome to be parted in two and in line with the observed nuclear-cytoplasmic distribution of the lncRNA. The co-purification of splicing factors might be predominantly of nuclear origin. We also selected numerous cytoplasmic RNP granule proteins and, among those, the significant enrichment of G3BP1, together with well-known G3BP1 interacting partners including UPF1 [34], was prevalent. Strikingly, anti-G3BP1 immunoprecipitation from cytoplasmic fractions followed by mass spectrometry uncovered a protein overlap of 40% with RNA-based *CALA* affinity selections, describing *CALA* to be part of cytoplasmic G3BP1-positive complexes. Trying to elucidate the function of *CALA* within those G3BP1-positive cytoplasmic RNPs, we found that *CALA* silencing leads to a tremendous loss of protein factors, suggesting *CALA* is of structural importance for cytoplasmic G3BP1-positive complexes.

G3BP1, the main cytoplasmic protein factor enriched by *CALA* affinity selections, has been well studied regarding its role in stress granule formation [32]. However, recent studies also proposed diverse functions for G3BP1 in the absence of cellular stress. For example, G3BP1 plays an important role in RNA metabolism via its endoribonuclease activity [54] and contributes to structure-mediated RNA decay [14]. Our mass spectrometry analysis revealed G3BP1 to interact with several proteins implicated in NMD in a *CALA*-dependent manner, as we found UPF1 and MOV10 to be reduced in their interaction with G3BP1 upon *CALA* silencing. Both factors are indispensable for efficient RNA decay [8,36]. In agreement, we show a direct implication of *CALA* in NMD regulation, as silencing of *CALA* leads to a significant increase in NMD target expression, but, at the same time, to no increase in precursor transcript expression.

Different lncRNAs have been shown to interact with G3BP1 [55,56], UPF1 [37,57,58,59] and MOV10 [37]. In contrast to these studies, which focus on, and elucidate, the individual interaction case-specifically, we, for the first time, have reported on a lncRNA to control mRNA turnover by being a crucial structural component of cytoplasmic G3BP1-positive RNPs (Figure 4). These results are in line with the ability of lncRNAs to act as complex-stabilizing scaffolds, as shown for the abundant lncRNA *NEAT1* in the context of paraspeckle formation [60]. Of note, the initial identification of *CALA* in different cell types [22] along with the ubiquitously expressed *CALA* binding proteins, strongly argues for a regulatory network not restricted to the endothelium. For example, *CALA* is additionally well expressed in diverse cancer types, conditions in which NMD is known to play a complex role [61]. Finally, the transcriptional regulation of *CALA* is not characterized. An *in-silico* promotor analysis suggests the involvement of the stress-sensitive transcriptional repressor CTCF [62], which is in line with our observed upregulation of *CALA* in different cellular stress settings. Taken as a whole, our study demonstrates the lncRNA *CALA* ensures the integrity and functionality of cytoplasmic G3BP1-positive complexes. By doing so, *CALA* regulates homeostatic RNA decay and gene expression, via a novel, lncRNA-dependent contribution to cytoplasmic decay pathways.

## 5. Limitations of the Study

The here presented work aimed to address the cytoplasmic role of G3BP1 and *CALA* in homeostatic RNA decay, focusing on its mechanism of action. RNA decay, however, is also coupled with stress granule formation. While initial studies done by us suggest that *CALA* silencing has no direct effect on stress granule formation, we cannot exclude an additional function of *CALA* under stress conditions. Future studies may, therefore, focus on the role of the identified *CALA*-G3BP1 interaction beyond homeostasis and comprehensively characterize and describe the target transcripts regulated by *CALA*-G3BP1 RNPs.

## Figures and Tables

**Figure 1 ncrna-08-00049-f001:**
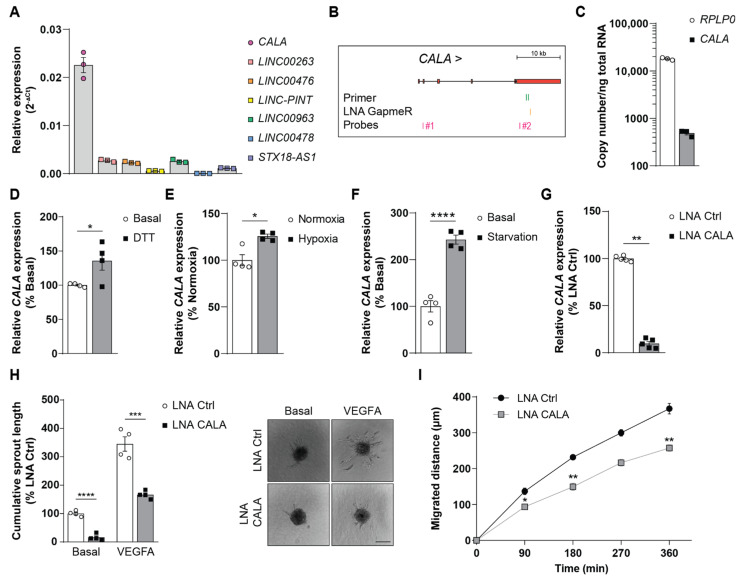
Stimuli-responsive lncRNA *CALA* regulates endothelial sprouting and migration. (**A**) Relative expression of *CALA* and indicated lncRNAs in HUVECs determined by RT-qPCR (n = 3). (**B**) Scheme of *CALA* highlighting primers, LNA GapmeR, and probes used. (**C**) Absolute quantification of *CALA* in HUVECs by digital RT-PCR (n = 3). (**D**–**F**) Relative *CALA* expression in HUVECs upon (**D**) ER stress (2 h, 1 mM DTT, n = 4), (**E**) hypoxia (24 h, 0.2% O_2_, n = 4), (**F**) starvation (reduced-serum media, 24 h, n = 4) compared to respective controls. (**G**) *CALA* expression after LNA-mediated silencing in HUVECs determined by RT-qPCR (n = 5). (**H**) Cumulative in vitro sprouting length of control and *CALA*-silenced HUVECs under basal conditions and VEGFA stimulation (n = 4). Representative pictures shown (scale bar = 100 µm). (**I**) Migratory capacity of HUVECs upon *CALA* silencing compared to controls (n = 3). Data information: In (**A**,**C**–**I**) data are represented as means ± SEM and derive from independent biological replicates, * *p* < 0.05, ** *p* < 0.01, *** *p* < 0.001, **** *p* < 0.0001. (**D**,**F**,**G**,**I**): two-tailed unpaired *t*-test, (**E**): Mann-Whitney U test, (**H**): two-way ANOVA.

**Figure 2 ncrna-08-00049-f002:**
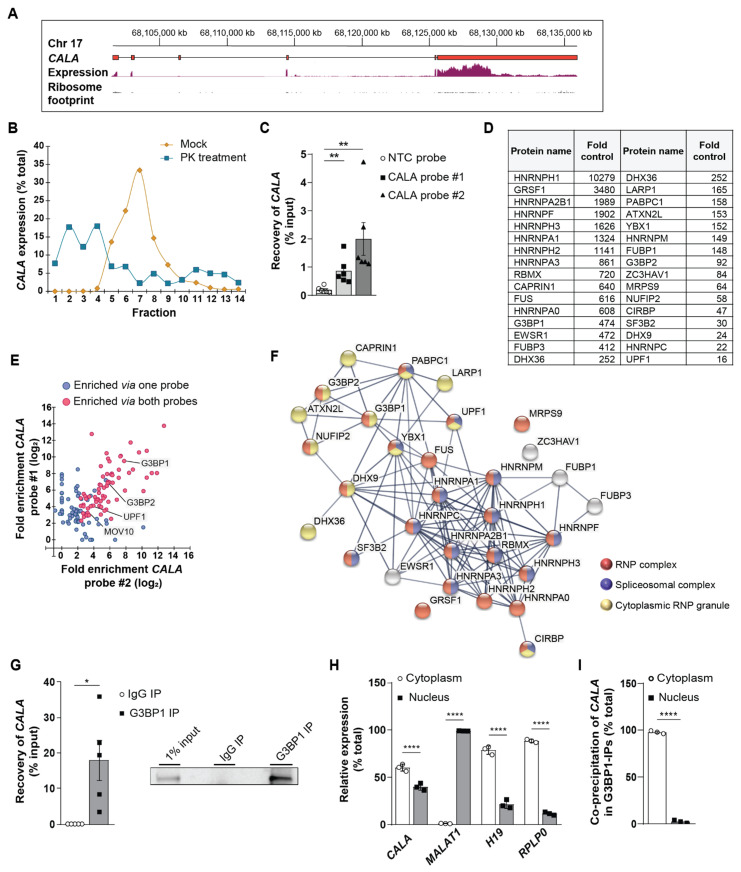
*CALA* interacts with multiple RNPs and primarily associates with cytoplasmic G3BP1. (**A**) *CALA* gene locus showing ribosome footprinting (GWIPs-viz-riboseq tracks) and aortic expression data (GTEx V8 RNA-seq). (**B**) Separation and detection of *CALA*-complexes by sucrose density gradient ultracentrifugation and RT-qPCR, comparing mock and proteinase K-treated whole HUVEC cell lysates (n = 1). Fractions 1 and 14 are top and bottom of the gradient, respectively. (**C**) Recovery of *CALA* from HeLa cell lysates in antisense affinity selections analyzed by RT-qPCR (n = 6). (**D**) Significantly enriched *CALA* protein interaction partners identified by mass spectrometry (≥10 unique peptides) of antisense affinity selection samples. (**E**) Significantly enriched proteins by either one (blue) or both *CALA*-targeting probes (pink) in antisense affinity selections identified by mass spectrometry (n = 6). (**F**) STRING analysis of significantly enriched proteins via both probes. (**G**) Validation of the *CALA*-G3BP1 interaction in whole HUVEC cell lysates by anti-G3BP1 RNA immunoprecipitations (RIPs) followed by RT-qPCR (n = 5). Representative western blot shown. (**H**) Subcellular localization of *CALA*, *MALAT1*, *H19*, *RPLP0* in HUVECs determined by RT-qPCR (n = 3). (**I**) Co-precipitation of *CALA* from anti-G3BP1 RIPs in cytoplasmic and nuclear HUVEC cell fractions assayed by RT-qPCR (n = 3). Data information: In (**C**,**G**–**I**) data are represented as means ± SEM and derive from independent biological replicates, * *p* < 0.05, ** *p* < 0.01, **** *p* < 0.0001. (**C**): two-tailed unpaired *t*-test (probe#1), Mann-Whitney U test (probe #2), (**G**–**I**): two-tailed unpaired *t*-test, (**H**): two-way ANOVA.

**Figure 3 ncrna-08-00049-f003:**
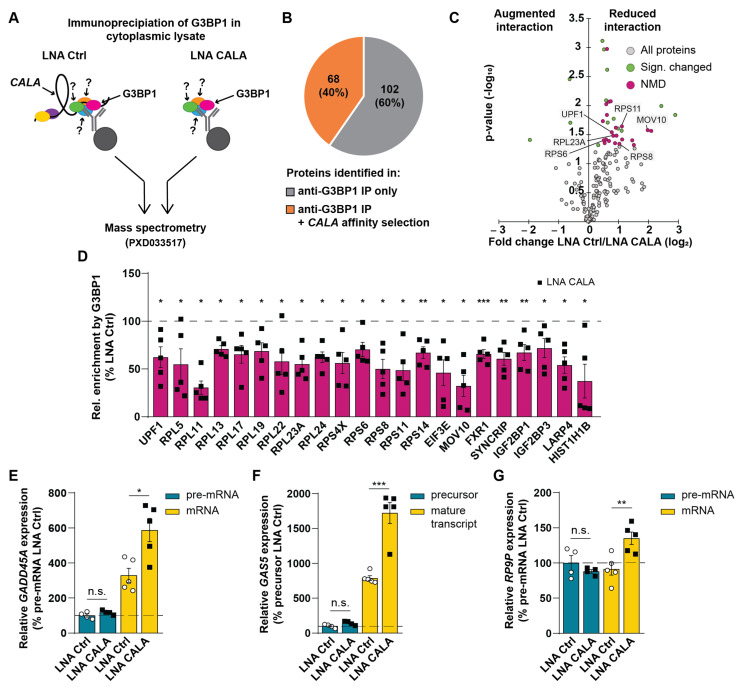
*CALA* impacts the composition and integrity of cytoplasmic G3BP1-RNPs driving mRNA decay. (**A**) Schematic representation of *CALA*-G3BP1-complex identification by anti-G3BP1 immunoprecipitations and mass spectrometry in control and *CALA*-silenced HUVECs. (**B**) Proportion of proteins identified exclusively in anti-G3BP1 IPs and overlapping with *CALA* antisense affinity selections. (**C**) G3BP1-co-immunoprecipitated proteins upon *CALA* silencing, discriminating augmented and reduced interactions. Nonsense-mediated decay (NMD) associated proteins were identified using Metascape. (**D**) Relative association of NMD proteins with G3BP1 in *CALA*-silenced HUVECs (n = 5). (**E**–**G**) Relative expression of (**E**) *GADD45A*, (**F**) *GAS5*, and (**G**) *RP9P* pre-mRNA and mRNA upon silencing of *CALA* determined by RT-qPCR (n = 5). Data information: In (**G**) data are represented as means ± SEM and derive from independent biological replicates, * *p* < 0.05, ** *p* < 0.01, *** *p* < 0.001, n.s.: non-significant. (**C**–**E**,**G**): two-tailed unpaired *t*-test, (**D**,**F**): Mann-Whitney U test.

**Figure 4 ncrna-08-00049-f004:**
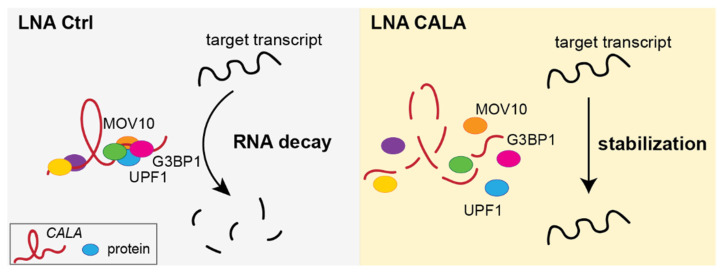
*CALA* promotes mRNA decay via cytoplasmic G3BP1-positive complexes. Model of *CALA* stabilizing cytoplasmic G3BP1-positive complexes, eventually enabling mRNA decay. Silencing of *CALA* disrupts the cytoplasmic interaction of G3BP1 with NMD regulatory factors including UPF1 and MOV10, ultimately compromising the decay of mRNAs.

## Data Availability

The mass spectrometry data sets have been deposited with the ProteomeXchange Consortium via the PRIDE partner repository and are publicly available with the data set identifiers PXD033516 and PXD033517.

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
