# Peer review of "The G3BP1-UPF1-Associated Long Non-Coding RNA CALA Regulates RNA Turnover in the Cytoplasm"

_ncrna, 2022, doi:10.3390/ncrna8040049_

Round 1

Reviewer 1 Report

Kirchhof et al. examine the interactions and stress-related roles of the long non-coding RNA CALA. The study reports CALA binding with G3BP1 and other RNA binding proteins, which in turn affects the assembly of UPF1 and other nonsense mediated decay factors. Overall, the reported biochemical interrogations are carefully done. A few additional pieces of evidence may strengthen the proposed mechanisms.

1.     All functional results for CALA currently come from 1 LNA sequence. An additional LNA, or siRNA, or knockout cell line might give more confidence that the effects observed are not due to off-target effects on other RNA species.

2.     As the study currently stands, it’s unclear if CALA is specifically suited to recruiting G3BP/UPF1/etc, or if any untranslated RNA is capable of contributing to RNP macro-assembly and stress granule formation. Referring to its role as “essential” may be an overstatement. The authors should demonstrate at minimum that LNA against a roughly similarly expressed lncRNA does not have the same effects. Alternatively, any insight into the sequences and motifs required for the interaction/effects could also strengthen the important role of CALA.

Minor comments:

1.     All qPCR results are RPLP0-normalized. Measuring another housekeeping gene would add further confidence that RPLP0 indeed does not change during the various stresses tested.

2.     Natural unmutated targets of NMD should be introduced early (line 50 is hinting at this, but can be more direct)

Reviewer 2 Report

In the manuscript by Kirchhof et al., the authors identified the lncRNA CALA as a regulator of RNA turnover in endothelial cell functions by forming cytoplasmic ribonucleoprotein complexes with G3BP1, UPF1 and other proteins identified by mass spectrometry. The authors further demonstrated that CALA silencing impaired degradation of NMD target transcripts. The present study is well designed, and the manuscript is well written, nevertheless, there remain several concerns.

Major:

1.     The authors purified the CALA-associated proteome using two antisense probes. It’s not surprising that a lot of RBPs would be identified to bind to such a long RNA. Some of these RBPs may be also associated with other RNA and RBPs, rendering it a very complicated complex. In this case, how can the authors prove which RBPs directly bind to CALA but are not a side-product of the enrichment? Because this is critical to informing the functions of CALA.

2.     Lines 393-395, the authors mentioned that “non-mutated GADD45A and GAS5 transcripts are reported to be key targets of NMD, as their RNA levels must be kept low for preventing apoptosis and growth arrest”. As shown in Figure 3E-F, GADD45A and GAS5 transcript expression levels were significantly increased by CALA silencing, however, in lines 259-261, the authors mentioned “While transfection of HUVECs with CALA-targeting LNAs led to a ~90% decrease in CALA levels (Figure 1G), we did not observe effects on cell proliferation, apoptosis, or endothelial permeability”. Can the authors explain why CALA silencing did not affect cell growth while the RNA levels of GADD45A and GAS5 were induced?

3.     Figure 3 showed the effect of CALA silencing on the G3BP1-associated proteome and even on mRNA decay. Can the authors further confirm the effect by overexpressing CALA and likewise, will overexpression after silencing counteract such an effect?

Minor:

1.     Line 330, the authors “hypothesize that CALA might be a central component of a multivalent RNP …” and then performed anti-G3BP1 RIP assays to test this hypothesis. The results indeed showed an association between G3BP1 and CALA, directly or indirectly, however, are not enough to tell that CALA is a “central” component for assembling proteins in an RNP. Again, the RIP assays can enrich all RNAs, including but not limited to CALA, and other proteins that are natively associated with the target G3BP1. So, the authors should either provide further evidence for their hypothesis or use other words for a more precise description based on the current results.

2.     Figure 2E and Figure 3C, it would be clearer to place beside some dots the names of the proteins that are particularly interesting and further analyzed in other panels.

3.     Line 398, change the expression to “In summary, our data indicate that CALA is required for the stabilization of …” because no evidence shows that “CALA generates a platform”.

Reviewer 3 Report

Essential role of lncRNA CALA in controlling cytoplasmic RNA 2 turnover” by Luisa Kirchhof et al describes the CALA, a lncRNA, regulates homeostatic RNA decay and gene expression. The lncRNA CALA ensures the integrity and functionality of cytoplasmic G3BP1-positive complexes. The paper is very well written in general. Moreover, several minor findings should be pointed out before acceptance.

Minor

1.     Please discuss how stress, like hypoxia or starvation condition, stimulates the expression of CALA?

2.     How about the expression of CALA in cancers?
